



# 1 Burn severity and vegetation type control phosphorus
# 2 concentration, molecular composition, and mobilization

Morgan E. Barnes[1], J. Alan Roebuck, Jr[2], Samantha Grieger[2], Paul J. Aronstein[3], Vanessa A.
Garayburu-Caruso[1], Kathleen Munson[2], Robert P. Young[1], Kevin D. Bladon[4], John D. Bailey[4],
Emily B. Graham[1,5], Lupita Renteria[1], Peggy A. O'Day[3], Timothy D. Scheibe[1], Allison N.
Myers-Pigg[2,6]
1 Pacific Northwest National Laboratory, Richland, WA, USA
2 Pacific Northwest National Laboratory, Sequim, WA, USA
3 Environmental Systems, University of California - Merced, Merced, CA, USA
4 College of Forestry, Oregon State University, Corvallis, OR, USA
5 School of Biological Sciences, Washington State University, Pullman, WA, USA
6 Department of Environmental Sciences, University of Toledo, Toledo, OH, USA
*Correspondence to*:
Allison Myers-Pigg (allison.myers-pigg@pnnl.gov)
Morgan Barnes (morgan.barnes@pnnl.gov)
Present Addresses:
Robert P. Young – Washington River Protection Solutions, P.O. Box 850 MSIN M0-01,
Richland, WA 99354
**Keywords:** char; fire impact; $^{31}$P NMR; P XANES; sagebrush shrubland; Douglas-fir forest;
soil organic matter; nutrient release
**Abstract**
Shifting phosphorus (P) dynamics after wildfires can have cascading impacts from terrestrial to
aquatic environments. However, it is unclear if post-fire responses are primarily driven by
changes to the molecular composition of the charred material or from the transport of P-
containing compounds. We used laboratory leaching experiments of Douglas-fir forest and
sagebrush shrubland chars to examine how the potential mobility of P compounds is influenced
by different burn severities. Burning produced a 6.9- and 29- fold increase in particulate P
mobilization, but a 3.8- and 30.5- fold decrease in aqueous P released for Douglas-fir forest and
sagebrush shrubland, respectively. P compound mobilization in the particulate phase was
controlled by solid char total P concentrations while the aqueous phase was driven by solubility
changes of molecular species. Nuclear magnetic resonance and X-ray absorption near edge
structure on the solid chars indicated that organic orthophosphate monoester and diester species
were thermally mineralized to inorganic P moieties with burning in both vegetation types. This
coincided with the production of calcium- and magnesium-bound inorganic P compounds. With
increasing burn severity there were systematic shifts in P concentration and composition—
higher severity chars mobilized P compounds in the particulate phase, although the magnitude of
change was vegetation specific. Our results indicate a post-fire transformation to both the
composition of the solid charred material and to how P compounds are mobilized, which may
influence its environmental cycling and fate.



**Short Summary**

Wildfires impact nutrient cycles on land and in water. We used burning experiments to
understand the types of phosphorous (P), an essential nutrient, that might be released to the
environment after different types of fires. We found that the amount of P moving through the
environment post-fire is dependent on the type of vegetation and degree of burning which may
influence when and where this material is processed or stored.

## 1 Introduction

Wildfires are a major modifier of the terrestrial landscape, directly burning around 4% of the
Earth's surface each year (Randerson et al., 2012). They affect both the terrestrial and adjacent
aquatic environments and, as such, are considered one of the largest drivers of aquatic
impairment (Ball et al., 2021). Organic and inorganic nutrient pools and fluxes can be altered by
burning through the loss of volatile compounds, changing physiochemical properties from the
incomplete combustion of organic material (from partially charred biomass to ash; collectively
referred to as chars (Bird et al., 2015)), and enhancing transport of materials from leaching and
erosion (Bodí et al., 2014). Movement of wildfire-derived material from terrestrial landscapes to
rivers has impacted 11% of total western United States river length in recent years (Ball et al.,
2021). As fire frequency, intensity, severity, and total area burned are expected to increase in
many regions, such as the western United States (Doerr and Santín, 2016; Haugo et al., 2019;
Jolly et al., 2015), it is important to understand the mechanisms behind how wildfires alter
nutrient quantity, composition, and mobilization.

Phosphorus (P; occurring primarily as orthophosphate $H_2PO_4^-$, $HPO_4^{2-}$, or $PO_4^{3-}$) is an essential
element (Smil, 2000) and is often a limiting nutrient to productivity in terrestrial and aquatic
environments (Elser et al., 2007). Ecosystem responses post-fire can include shifting terrestrial
nutrient acquisition by decreasing phosphatase activity and promoting net primary production
(Dijkstra and Adams, 2015; Saa et al., 1993; Vega et al., 2013). Phosphorus-containing
compounds transported to aquatic environments can also increase aquatic productivity,
influencing invertebrate and fish size and growth rate (Silins et al., 2014). While there is largely
an agreement across studies that P becomes enriched in chars after wildfire (Butler et al., 2018;
Elliott et al., 2013; García-Oliva et al., 2018; Schaller et al., 2015), with increased concentrations
in mineral soil (Butler et al., 2018) and river systems following wildfire (Lane et al., 2008;
Mishra et al., 2021; Rust et al., 2018), we are lacking a systematic understanding on how
variable burning conditions mediate the P concentration of charred organic material, and the role
of different fire-prone vegetation types (but see (Schaller et al., 2015; Wu et al., 2023b;
Yusiharni and Gilkes, 2012)) on availability for mobilization. Prescribed burns and wildfires
occur across a range of burning conditions (Merino et al., 2019; Santín et al., 2018; Vega et al.,
2013), which results in a mosaic of post-fire ecosystem responses on the landscape (Keeley,
2009). Therefore, understanding how P biogeochemistry is altered along a burn gradient will
provide insights on heterogenous responses observed across burned landscapes.



In the environment, P is found in multiple molecular moieties (i.e., orthophosphate, phosphonate,
orthophosphate monoester, orthophosphate diester, polyphosphate; orthophosphate monoester
and orthophosphate diester compound classes, referred to as the ester bonds moving forward)
which exist in different chemical states (i.e., adsorbed on surfaces, incorporated into minerals,
precipitated with metals). Chemical speciation influences the solubility and mobility of P, which
in turn impacts its bioavailability (Li and Brett, 2013; Turner et al., 2003b; Weihrauch and Opp,
2018; Yan et al., 2023). For example, bonding energy, or strength of the bonds, of the chemical
species generally increases from organic P to sorbed and mineral bound P species (Weihrauch
and Opp, 2018). The fate of these P species is determined by biological, chemical, physical, and
environmental factors, which vary in space and time (Condron et al., 2015; Yan et al., 2023).
Thus, the potential influence of wildfire effects on P dynamics and ecosystem productivity
cannot be adequately ascertained by only characterizing P concentration. Compared to changes
in total P concentration, there is less understanding of P molecular composition in charred
material and the impact this has on its mobilization (Robinson et al., 2018; Wu et al., 2023a). As
such, it is unclear if P biogeochemical responses post-fire are due to changing composition of the
charred material (i.e., composition controlled) and/or an artifact of how P compounds are
transported (i.e., mobilized from the solid char to then be transported through the environment).
Recent research on laboratory-produced plant-derived chars has demonstrated the use of NMR to
quantify P moiety (Sun et al., 2018; Uchimiya and Hiradate, 2014; Wu et al., 2023b; Xu et al.,
2016; Yu et al., 2023) and XANES to identify chemical state (Robinson et al., 2018; Rose et al.,
2019; Wu et al., 2023a; Yu et al., 2023). Taken together, these complementary techniques are
useful tools to provide a holistic understanding of P molecular composition and can help to
determine the environmental fate, as certain compounds are preferentially volatilized, produced,
and transported across the landscape (Son et al., 2015).
Vegetation burn severity, a common metric to describe how wildfires impact ecosystems, allows
for a post-fire assessment of ecosystem impacts (Keeley, 2009). Burn severity is determined by
the extent of organic matter loss or change after fire and is influenced by fire intensity, heating
duration, degree of live or dead plant material, and fuel moisture, among other factors (Keeley,
2009). However, relatively few studies relate burn severity to fire effects on P biogeochemistry
(Souza-Alonso et al., 2024; Vega et al., 2013) even though it is a more commonly used field
metric than fire intensity because it can be measured after the burn (Zavala et al., 2014). Thus,
burn severity allows for understanding how burning conditions beyond temperature influence
ecosystems. Experimental studies along burn severity gradients provide an opportunity to better
understand field conditions post-fire. To understand the amount and types of materials that could
be transported along a burned gradient, we examined how P concentration and molecular
composition in solid chars and their leachates vary across a burn severity gradient. We
hypothesize that changing P composition in the solid charred materials with increasing burn
severity will influence the leachability of P compounds in the particulate and aqueous phases,
and this will be moderated by vegetation type. To test this hypothesis and better understand the
amount and types of materials that could be mobilized along a burned gradient, we examined
how burn severity influences P concentration and molecular composition in experimentally
generated solid chars and their leachates.
**2 Materials and Methods**



All datasets and detailed methodology used in this manuscript are available from Grieger et al.
(Grieger et al., 2022) version 3 and Barnes et al. (in prep) on the Environmental System Science
Data Infrastructure for a Virtual Ecosystem (ESS-DIVE) repository.
*2.1 Burn Experiments*
Vegetation was collected from two fire-prone landscapes of contrasting vegetation types from
the Pacific Northwest, USA that also have differing wildfire characteristics (Roebuck et al.,
2024). We represented these landscapes by collecting the dominant vegetation present. In this
study, we chose to explore Douglas-fir forests (*Pseudotsuga menziesii*), which tend to burn at
higher intensities given fuel loading, and sagebrush shrublands (*Artemisia tridentata*), which
tend to burn at lower intensities (Stavi, 2019). In addition, fire exclusion has resulted in Douglas-
fir forest encroachment into historically sagebrush shrubland habitat, altering fire dynamics of
these landscapes (Everett et al., 2000; Heyerdahl et al., 2006; Strand et al., 2013). Samples were
chosen to be representative of possible living vegetation and litter materials of the dominant
species from these landscapes. For Douglas-fir, a mix of living and dead material was collected,
while sagebrush was in partial senescence upon collection. Woody and canopy materials were
mixed at a known ratio before each burn, and this was held constant for each burn (Grieger et al.,
147  2022).

Chars were generated using an open air burn table, as biochars produced in laboratories have
been found to be compositionally different than chars generated in open air burns and wildfires
(Myers-Pigg et al., 2024; Santín et al., 2017). To create burns that would result in a range of
vegetation burn severities, we manipulated fire behavior on the burn tables by varying burn
temperature, duration of heating, fuel moisture content, fuel density, and vegetation status (i.e.,
living or litter). Thermocouples were used to monitor temperature over the burn duration, and
char grab samples were targeted for 300 °C, 600 °C, and when flames and smoldering
commenced (sagebrush shrubland burns did not reach 600 °C). Char burn severity was classified
following US Forest Service field metrics based on ash color, degree of consumption, and degree
of char (Grieger et al., 2022; Parsons et al., 2010) (Fig. S1). Thus, burn severity was determined
by the extent of organic matter loss or change after fire and is influenced by fire intensity,
heating duration, degree of live or dead plant material, and fuel moisture, among other factors
(Keeley, 2009). Unburned samples and chars were air dried; subsamples were finely ground for
elemental composition, and were stored in the dark at room temperature until further analysis.
*2.2 Elemental Analysis of Solid Samples*
Total P, sulfur (S), aluminum (Al), iron (Fe), magnesium (Mg), calcium (Ca), sodium (Na), and
potassium (K) were measured using an inductively coupled plasma optical emission
spectrometer (ICP-OES) model Optima 7300 DV (PerkinElmer, Waltham, MA).  Solid samples
were digested with aqua regia at 130 °C for 8 h in an incubation oven (ThermoFisher Scientific,
Waltham, MA).
For samples that underwent NMR analysis, approximately 0.5 g of finely ground sample was
extracted in a 10 mL solution of 0.25 M NaOH and 0.05 M EDTA for 16 h, followed by
centrifugation, filtration, and measurement on ICP-OES (Sun et al., 2018; Turner et al., 2003b).
The goal of the NaOH-EDTA extraction is to get the maximum amount of P into solution.
Extraction efficiencies are reported in Table S1 (see SI section Method Limitations for additional
information).



*2.3 Solution $^{31}P$ NMR on Solid Samples*
After aliquoting 3 mL of the NaOH-EDTA extracts for ICP-OES, the remaining supernatants
were frozen and lyophilized to concentrate the extracted compounds. Immediately prior to
running NMR experiments (Environmental Molecular Science Laboratory; EMSL, Richland,
WA), freeze-dried extracts were reconstituted in 0.52 mL deuterium oxide ($D_2O$) and 0.26 mL of
10 M NaOH, and 0.52 mL of a solution containing 0.5 M NaOH and 0.1 M EDTA. Full
experimental $^{31}P$ NMR measurement details are provided in the supporting information. In brief,
NMR measurements were conducted on an Agilent DD2 spectrometer operating at a field
strength of 14.1T (242.95 MHz $^{31}P$), equipped with a 5mm Varian broadband direct detect probe.
Experiments were conducted at a regulated temperature of 20.0°C. A standard 1D pulse and
acquire experiment was performed using a 90° pulse width and recycle delay equal to $5 \times T1$,
which were calibrated and measured individually for each sample using the orthophosphate peak
present in each. Samples were measured for 16 h each with the number of transients acquired
dependent upon T1 for each individual sample. Post-acquisition processing and analysis was
performed using Mnova 14.0.1 (Mestrelab Research, Spain). Details regarding classification of
major P forms, identification of specific P compounds from spiking experiments, quantitation,
and method limitations are described in detail in the supporting information (Cade-Menun, 2015;
Doolette et al., 2009; Recena et al., 2018) (Fig. S2).
*2.4 Solid Sample P XANES*
X-ray absorption near edge structure (XANES) is a complementary technique to solution $^{31}P$
NMR because it can discern the complexation environment of P in solid samples (see SI section
Method Limitations for additional information). Bulk XANES was conducted on beamline 14-3
at the Stanford Synchrotron Radiation Lightsource (SSRL, Stanford, CA). The beamline was
calibrated at the P K-edge with the first peak of tetraphenylphosphonium bromide at 2146.96
eV.
Sample spectra were fit using least-squares linear combination in Athena (Ravel and Newville,
2005) (Fig. S3). Baseline correction and edge-step normalization parameters were varied for
individual samples and reference compounds to reduce error (Werner and Prietzel, 2015). Fits
were performed with the component sum not forced to unity, a maximum of three reference
compounds, and only fits within $\pm 2.5\%$ were used. If a component fit $< 5\%$, then this reference
compound was removed, and the sample was refit with the remaining compounds. The R-factor
of all sample fits were $< 0.05$ (Table S2), indicating a good quality of fit (Kelly et al., 2015). Fits
were performed with a variety of Ca, Al, Fe, Mn, K, and Na inorganic and organic P-containing
reference compounds. Individual inorganic P compounds ($P_i$; includes phosphate and
pyrophosphate moieties) reference compounds were grouped based on the associated metal and
all organic P compounds ($P_o$; includes monoester and diester moieties) were kept as a separate
category (Fig. S3; Table S3). Additional information on sample preparation, linear combination
fits, reference compounds, and method limitations are described in the supplemental information
(XANES Methodology section).
*2.5 Leaching Experiments*
Leachates from unburned material and char samples were generated in triplicate. Briefly, 25 g of
unground sample was shaken in the dark for 24 h in 1000 mL of synthetic rainwater (pH ~ 5) to
simulate what might be mobilized by rain events from the solid material and subsequently



transported from terrestrial to aquatic environments (Grieger et al., 2022). Our starting mass was
kept constant to understand differences in the amounts of materials leached across burn severity
gradients, and so our results are directly comparable to temperature gradient studies (Bostick et
al., 2018). Therefore, leaching experiments had a different goal of simulating natural
mobilization of P compared to the NMR extractions, where we tried to maximize P extracted.
Leachates were filtered through a PTFE mesh (2 mm x 0.6 mm) followed by a pre-combusted
GF/F filter (< 0.7 μm). Aliquots were immediately taken for subsequent analysis and preserved
according to analytical needs described below.

*2.6 Elemental Analysis of Leachates*

Coarse filtered (< 2 mm) and < 0.7 μm filtered (i.e., aqueous phase) leachates were preserved in
1% nitric acid and stored at 4 °C until analysis. Aliquots of 5 mL were transferred to 15 mL
centrifuge tubes, acidified to 10% (v/v) trace metal grade hydrochloric acid and 4% (v/v) trace
metal grade nitric acid. Tubes were fully sealed and heated at 85 °C for 2.5 h in an incubation
oven (ThermoFisher Scientific, Waltham, MA) and then total elemental analysis were measured
by ICP-OES. Total P of the leachate particulate phase (2 mm to 0.7 μm) was calculated as the
difference between the coarse filtered and aqueous phase.
Molybdate reactive P was determined on aqueous phase leachate aliquots preserved in 0.2%
sulfuric acid and stored at 20 °C, following EPA method 365.3 (Method 365.3: Phosphorus, All
Forms (Colorimetric, Ascorbic Acid, Two Reagent)). Aqueous non-molybdate reactive P was
calculated as the difference between aqueous total P (as measured by ICP-OES) and molybdate
reactive P.

*2.7 Data Analyses*

Leachable P (mg g P$^{-1}$; particulate and aqueous phases separately) was calculated by normalizing
to the P concentration of the solid samples following Equation 1 (Fischer et al., 2023):

$$Leachable\ P_{particulate\ or\ aqueous} = \frac{leachate\ P\ (mg\ L^{-1})\ x\ leaching\ volume\ (L)}{mass\ of\ dry\ char\ (g)\ x\ P\ content\ of\ dry\ char\ (mg\ P\ g^{-1})}$$

All statistical tests were conducted in R version 4.2.3 (R Core Team, 2023). Data calculations,
statistical analyses, and figures are available on Github ([https://github.com/river-corridors-](https://github.com/river-corridors-)
[sfa/rcsfa-RC3-BSLE_P](https://github.com/river-corridors-sfa/rcsfa-RC3-BSLE_P)). For all statistical analyses, model assumptions were assessed with a
Shapiro-Wilk test of normality using the package stats (R Core Team, 2023) and spread-location
plots to inspect homoscedasticity. All analyses met assumptions after log transformation.
Significance was determined at the α = 0.05 level. All data are reported as the mean ± standard
deviation unless otherwise stated.
Separate analysis of variance (ANOVA) models were used to test how burn severity, vegetation
type, and their interaction influences solid P concentration. For leachate samples (i.e., particulate
total P, aqueous total P, aqueous molybdate reactive P), mixed-effect models were run with the
same fixed effects as the solid samples and a random effect was used to account for triplicate
leachates produced from the same solid sample. Mixed effect models were performed with the
lme4 package (Bates et al., 2015) and were fit by maximum likelihood. Variance Inflation
Factors were used to inspect for multi-collinearity of fixed effects with the car package (Fox and
Weisberg, 2018). Post-hoc pairwise comparisons were conducted using least squares means in
the emmeans package (Lenth, 2023).



Path analysis was conducted to analyze the hypothesized relationships that may explain how
burn severity and vegetation type influence P compound mobilization (i.e., leachable particulate
or aqueous phase P concentration) indirectly through changes in char conditions (i.e., P
concentration and chemical composition). Calcium-bound $P_i$ was used as a proxy for chemical
composition because it is a primary control of P compound solubility in charred materials
(Schaller et al., 2015; Uchimiya and Hiradate, 2014; Wu et al., 2023b; Yu et al., 2023).
Phosphorus compound mobilization was estimated as the average leachable P from the parent
solid samples. Models were run with the sem package (Fox, 2006), with burn severity and
vegetation type directly impacting the P concentration and proportion of Ca- $P_i$ in the solid
samples, which in turn influence the leachable P concentration. Vegetation type is also set up to
directly impact burn severity (Fig. S4).
**3 Results and Discussion**
*3.1 The magnitude of char P increase with burn severity depends on vegetation type*
In our study, using experimental open air burns, we found total P concentration increased with
burn severity in both Douglas-fir forest and sagebrush shrubland solid samples (Fig. 1). Our
findings were consistent with observations of increasing P concentration from laboratory-
produced chars (García-Oliva et al., 2018; Zheng et al., 2013) and in chars collected shortly after
wildfire and prescribed burns (Butler et al., 2018). The P concentration in unburned Douglas-fir
forest samples was $1.3 \pm 0.5$ g P kg$^{-1}$ and increased to an average of $6.2 \pm 1.9$ g P kg$^{-1}$ in high-
severity burns (ANOVA post hoc $p < 0.001$), with temperatures that reached an average of $704 \pm$
78 °C. On the other hand, unburned sagebrush shrubland material contained 1.3 g P kg$^{-1}$
compared to $14.5 \pm 3.5$ g P kg$^{-1}$ in the moderate-severity burns (ANOVA post hoc $p < 0.001$)
that reached $530 \pm 25$ °C. The observed increase in char P indicated that retention (i.e.,
condensation) outweighed loss via volatilization. Generally, P and metal cations volatilize at
higher temperatures (>774 °C or greater) than carbon (C) and nitrogen (N) (>200 °C), so they are
often retained in charred material rather than lost in gaseous form (Son et al., 2015).






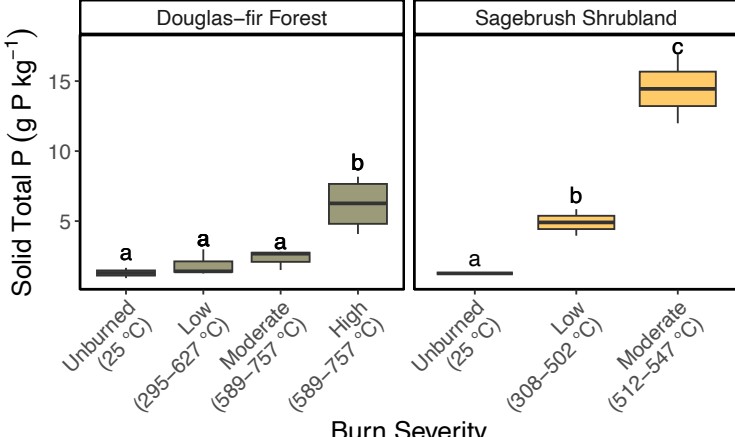

Figure 1. Phosphorus concentration (g P kg$^{-1}$) in the solid samples along Douglas-fir Forest and Sagebrush shrubland burn severity gradients. Ranges of maximum temperatures (°C) reached within a respective burn severity category are reported in parentheses. Letters denote post hoc findings of burn severity significant differences within a vegetation type.

Although P concentration in solid samples increased from unburned to the highest severity
classification reached in both vegetation types, the magnitude was vegetation dependent
(ANOVA interaction term: F = 6.23, $p$ = 0.014). In Douglas-fir forest chars, P concentration was
unchanged by burning until high-severity was reached (post hoc test; low: $p$ = 0.658, moderate: $p$
= 0.277, high: $p$ < 0.001), while P in sagebrush shrubland chars increased even after low-severity
burns (post hoc test; low: $p$ = 0.034, moderate: $p$ < 0.001). Post hoc tests further identified that
the P concentration of sagebrush shrubland chars was significantly greater than Douglas-fir
forest within the same burn severity classification (low: $p$ = 0.0038; moderate: $p$ < 0.001), even
though unburned samples were not statistically different ($p$ = 0.962). On average, total P in
sagebrush shrubland chars were 2.7 and 6.2 times higher than Douglas-fir forest in low and
moderate-severity burns, respectively (Fig. 1).
Remarkably, P in moderate-severity sagebrush shrubland chars was even higher than Douglas-fir
forest high-severity chars. Higher maximum char temperatures or burn duration does not explain
why P concentration is greater in burned sagebrush shrubland material compared to Douglas-fir
forest; sagebrush shrublands experienced lower temperatures (530 ± 25 °C) and burn duration
(202 ± 3 minutes) in moderate-severity burns compared to Douglas-fir forest high-severity burns
(704 ± 78 °C; 783 ± 195 minutes; Table S4).
One mechanism that could explain such results is that sagebrush shrublands may be composed of
volatile compounds that are more susceptible to loss compared to Douglas-fir forests, leading to
selective enrichment of P compounds relative to Douglas-fir forest chars. However, emission
factors and total volatile organic compounds from sagebrush and coniferous fuels are relatively
similar (Hatch et al., 2019; McMeeking et al., 2009). This suggests that the observed enrichment



of sagebrush shrubland P with burning may be due to differences in how specific P compounds
in the sagebrush shrubland materials are retained compared to other compounds that are
combusted efficiently in those chars, which can arise from different fire conditions (Fiddler et
al., 2024). Sagebrush shrublands may be more susceptible to changing P dynamics post-fire
because chars are likely enriched in P to a greater extent than Douglas-fir forests, even at low
severities.
*3.2 Solid char molecular composition is influenced by burn severity and vegetation type*
Organic P in the solid char was progressively transformed to inorganic species across both
vegetation types. Unburned Douglas-fir forest and sagebrush shrubland had similar initial
percentages of total organic P with $40.5 \pm 3.3\%$ and $53.7\%$, respectively (identified by NMR
extracts, Fig. 2; also supported by XANES on solid phase, Fig. 3). As burning progressed, the
total organic P pools reduced to only $12.6 \pm 8.2\%$ for Douglas-fir forest and $10.4 \pm 8.4\%$ for
sagebrush shrubland low-severity chars. While organic P moieties were still present in Douglas-
fir forest chars produced at moderate severities ($4.4 \pm 4.2\%$), <1% was measured in sagebrush
shrubland. Moderate-severity sagebrush shrubland chars more closely resembled high-severity
Douglas-fir forest with nearly all organic P moieties lost (<1%). This further supports the
conclusion that different fire conditions were experienced by Douglas-fir forest and sagebrush
shrubland in our simulated burns. Although it has been suggested that organic P can be fully
transformed to inorganic species at 200 °C (García-Oliva et al., 2018), another study of organic
horizons found organic P moieties persisted after low, moderate, and high-severity fires that
reached up to 872 °C (Merino et al., 2019). We measured organic P in burns that reached above
600 °C, suggesting that the thermal mineralization of organic to inorganic P compounds is
controlled by microscale differences in temperature and selective protection rather than what is
observed at overall bulk temperatures, and is likely a result of the interaction between
temperature, burn duration, and vegetation type experienced by these microsites (Galang et al.,
2010; Lopez et al., 2024).




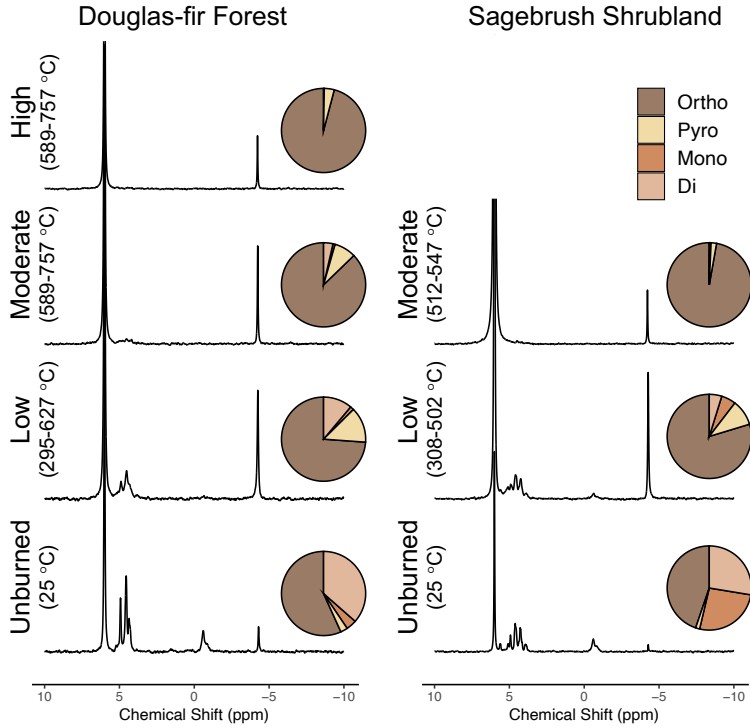

Figure 2. Solution $^{31}$P nuclear magnetic resonance (NMR) spectra from a representative solid char sample of each burn severity and vegetation type. The number of scans varied for each sample, based on relaxation time, and therefore direct comparisons of peak intensities can only be made within a spectrum (see additional details in SI NMR Methods). Averaged replicates are represented by pie charts for the proportions of orthophosphate (ortho), pyrophosphate (pyro) monoesters (mono) and diesters (di). Orthophosphate and pyrophosphate are inorganic species (brown colors) and monoester and diesters are organic species (orange colors). Ranges of maximum temperatures (°C) reached within a respective burn severity category are reported in parentheses. See SI sections NMR Methodology and Method Limitations for additional details.


Previous studies have suggested charred materials containing diester species (two C moieties per
P) are more vulnerable to thermal mineralization than monoesters (one C moiety per P) (García-
Oliva et al., 2018; Turrion et al., 2010). However, we found diester and monoester species
followed similar proportional decreases in our chars with burning (Fig. S5). Hence, both readily
available (i.e., diester) and less labile (i.e., monoester) organic P species (Condron et al., 2015)
were converted to inorganic P at comparable rates, which is similar to forest and shrubland



organic horizons subjected to prescribed fire (Merino et al., 2019). This suggests there is not a
fundamental molecular difference in how these moieties respond to burning in organic material,
but instead the preferential loss of diesters in burned mineral soil may be because the stronger
sorption of monoesters to soil particles attenuates the heat.

Because diester and monoester species were lost at similar proportions, the composition of the
unburned material dictated the resulting char P composition and potential bioavailability. Across
both vegetation types, we identified phospholipids, DNA, and RNA (diester region) and phytate
and sugar phosphates (monoester region; Fig. 2; Table S5), which follows other studies of
vegetation P composition (Doolette and Smernik, 2016; Noack et al., 2012). However, the
proportions of these species were vegetation dependent, where unburned Douglas-fir forest was
dominated by diesters ($36.5 \pm 9.1\%$) with minor percentages of monoesters ($4.1 \pm 5.7\%$),
whereas sagebrush shrubland was nearly equal parts diesters (27.6%) and monoesters (26.1%).
RNA, DNA, phospholipids, and sugar phosphates are considered bioavailable due to their weak
adsorption, whereas phytate strongly sorbs to both organic and inorganic particles making it
relatively less available for biological uptake (Condron et al., 2015; Li and Brett, 2013; Turner et
al., 2003a). Douglas-fir forest was composed of a greater proportion of these bioavailable
organic species in unburned ($36.8 \pm 7.6\%$) and low-severity burns ($12.4 \pm 8.4\%$) compared to
sagebrush shrubland (unburned: 32.4; low: $8.0 \pm 4.5\%$).

With increased burn severity, Douglas-fir forest (high-severity) and sagebrush shrubland
(moderate-severity) organic speciation converged with only < 1% of organic P (as RNA)
remaining. Prior studies using NMR in plant-based biochar produced from $300 - 800$ °C found
char was composed of entirely inorganic P, including orthophosphate (27–97%) and
pyrophosphate (3–71%; although one sample produced at 350 °C was 2% phospholipids) (Sun et
al., 2018; Uchimiya et al., 2015; Uchimiya and Hiradate, 2014). The unburned parent material in
these studies had variable starting compositions with organic P ranging from 3–87% (as phytate).
The extent of organic P loss in these studies is most similar to our higher severity samples, once
again demonstrating that more than temperature determines the composition of P in charred
material. Overall, these findings suggest organic P moieties in charred material are determined by
the degree of burning, where lower severity chars resemble the starting composition, and this is
influenced by vegetation type.

As organic species were thermally mineralized in our chars, inorganic P, such as pyrophosphate,
was produced (Fig. 2). Pyrophosphate is thought to largely originate from fungal tissue
(Bünemann et al., 2008; Makarov et al., 2005), and it has been found in some plants (Noack et
al., 2012; Wu et al., 2023b). We found pyrophosphate peaked in low-severity chars across both
vegetation types, reaching $13.6 \pm 3.1\%$ in Douglas-fir forest and $9.9 \pm 6.2\%$ in sagebrush
shrubland burns. Prior NMR studies on plant chars produced between 350 - 800 °C have also
observed an increase in the proportion of pyrophosphate relative to unburned material, followed
by a decrease at higher charring conditions (Sun et al., 2018; Uchimiya and Hiradate, 2014).
Variability in pyrophosphate from naturally produced chars has also been observed. For
example, post wildfire, pyrophosphate was ~3% in a pine forest (García-Oliva et al., 2018),
absent in a eucalyptus forest (Santín et al., 2018), 0–13% of cedar-hemlock forests (Cade-Menun
et al., 2000), and 3–7% from pine forests and shrublands (Merino et al., 2019). Thus burned
organic material, especially in chars produced at low-severity wildfire and prescribed burns, may
be an important, yet underappreciated, source of pyrophosphate in the environment.



The production of pyrophosphate in our charred plant material is likely a result of the initial
organic matter composition and burning conditions (Wu et al., 2023b; Yu et al., 2023).
Pyrophosphate and other polyphosphates can be produced from orthophosphate during burning,
with the thermal degradation of phytate (organic P; monoester) contributing more
orthophosphate (Robinson et al., 2018; Rose et al., 2019; Uchimiya and Hiradate, 2014).
Pyrophosphate was greater in Douglas-fir forest chars compared to sagebrush shrublands, even
though sagebrush shrubland chars contained more phytate in the unburned material (Fig. 2). This
indicates pyrophosphate was primarily produced from polymerization and dehydration of
orthophosphate, and not from thermal degradation of phytate in our chars (Uchimiya and
Hiradate, 2014).
Although pyrophosphate peaked in low-severity chars, we found the percentage of total
inorganic P species continued to increase with burning across both vegetation types, as measured
by NMR on solid extracts and XANES of intact solid samples (Fig. 2, Fig. 3; Tables S2 and S5),
demonstrating additional transformations to P composition with increasing severity. Inorganic
species, measured by XANES, in unburned material was composed largely of P compounds
associated with Fe (37% sagebrush shrubland; $40 \pm 5\%$ Douglas-fir forest; fitting primarily as $P_i$
sorbed to the surface of goethite) and a minor component of Ca-bound $P_i$ species ($3 \pm 3\%$
Douglas-fir forest; 9% sagebrush shrubland; fitting mostly as apatite). The proportion of Ca- and
Mg-$P_i$ (fitting as magnesium phosphate and/or struvite) increased with burn severity (Fig. 3;
Table S2). Douglas-fir forest high-severity chars had $52.8 \pm 8.3\%$ Ca-$P_i$ and $29.0 \pm 9.9\%$ Mg-$P_i$,
while sagebrush shrubland moderate-severity chars contained $45.1 \pm 0.1\%$ Ca-$P_i$ and $53.7 \pm$
$0.1\%$ Mg-$P_i$.
Other studies using XANES supports the production of Ca-$P_i$, along with Fe- or Mg-$P_i$ in plant-
based chars and ash (Robinson et al., 2018; Sun et al., 2018; Uchimiya and Hiradate, 2014; Wu
et al., 2023a), whereas studies using other techniques (solid-state NMR, sequential fractionation)
have found higher temperatures result in greater Ca- and Al-$P_i$ (García-Oliva et al., 2018; Xu et
al., 2016). Hydroxyapatite and other stable forms of Ca-$P_i$ minerals are known to be produced by
organic matter combustion (Uchimiya and Hiradate, 2014), so it follows that these P species are
produced with burning and progressively increase along our burn severity gradient. P compound
bonding environments have been found to resemble stoichiometric ratios of the burned material
(Wu et al., 2023a; Zwetsloot et al., 2015) .Our findings support this where Ca- and Mg-$P_i$ species
increased as the proportion of Ca and Mg also increased (Fig. 3; Tables S2, S6, and S7).
Phosphorus mobility and bioavailability of P compounds are likely influenced by increased
inorganic P proportions because Ca-$P_i$, especially apatite, is considered to have low water
extractability and apparent bioavailability (García-Oliva et al., 2018; Li and Brett, 2013;
Zwetsloot et al., 2015).






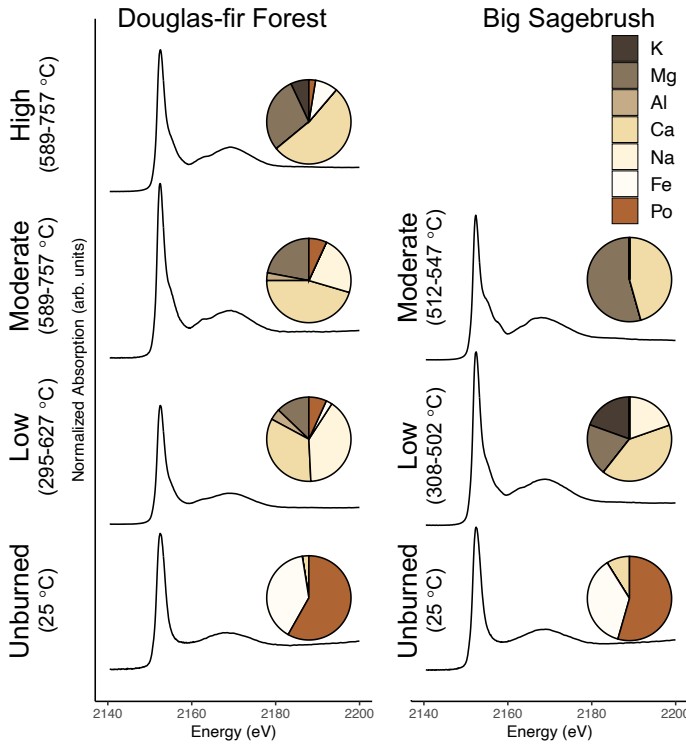

Figure 3. Phosphorus K-edge X-ray absorption near edge structure (XANES) spectroscopy from a representative solid unburned and char sample of each burn severity and vegetation type. Averaged replicates are represented by pie charts for the proportions of $P_i$ associated with K-, Mg-, Al-, Ca-, Na-, and Fe (brown colors) and $P_o$ species grouped together regardless of metal association (orange color; see SI XANES Methods for additional details). Ranges of maximum temperatures (°C) reached within a respective burn severity category are reported in parentheses. See SI sections XANES Methodology and Method Limitations for additional details.


*3.3 Leachable particulate- and aqueous-bound P have contrasting mobilization patterns with*
*burning and are under differing controls*
As burn severity increased, the enriched P of the solid chars resulted in greater particulate P
mobilized (assessed via leaching experiments), regardless of vegetation type (β = 0.78, *p* <
0.001, $r^2$ = 0.68; Fig. 4, Fig. 5). Burning resulted in a 6.9- and 29- fold increase of particulate P
mobilization from Douglas-fir forest (high-severity) and sagebrush shrubland (moderate-
severity) chars, respectively (Fig. 4). Phosphorus compounds may be largely physically protected
in the matrix of the charred material (70–90% residual P in sequential fractionation scheme (Wu



et al., 2023b)), therefore it follows that particulate P patterns are controlled by changes in solid
char concentration; charred material becomes enriched with P and there is production of highly
mobile particulates (such as ash (Blake et al., 2010)). Path analysis identified that burn severity
($\beta = 0.61$, $p < 0.001$) and vegetation type ($\beta = 0.65$, $p < 0.001$) had direct influence on solid char
P concentration ($r^2 = 0.64$; Fig. 5). Mixed effect model results further demonstrate that the effect
burn severity has on leachable particulate P is vegetation dependent (interaction term of mixed
effect model; $p = 0.009$). Moderate-severity sagebrush shrubland chars mobilized 5.2 times more
P in the particulate phase than Douglas-fir forest ($p = 0.04$). Particulate P mobilized from charred
material can be transported to waterways, as a meta-analysis found unfiltered P concentrations in
the western United States increased ~1.7 times after wildfire ($n = 46$) (Rust et al., 2018).
In contrast to leachable particulate P, mobilization of P in the aqueous phase decreased 3.8-fold
for Douglas-fir forest and 30.5-fold for sagebrush shrubland with burning (Fig. 4). Instead of
concentration-controlled like particulate P, aqueous P mobilization was composition-controlled,
represented as percentage of Ca-$P_i$ in our path analysis ($\beta = -0.44$, $p = 0.041$, $r^2 = 0.34$; Fig. 5).
Prior work from laboratory-produced plant chars have also found decreased water-soluble P even
though solid char concentration increased with burning (Gundale and DeLuca, 2006; Mukherjee
and Zimmerman, 2013; Wu et al., 2011; Yu et al., 2023; Zheng et al., 2013). Phosphorus
compound adsorption to multivalent cations ($Ca^{2+}$, $Mg^{2+}$, $Fe^{3+}$, and $Al^{3+}$) can decrease aqueous
phase export (Glaser et al., 2002). Indeed, we found higher severity burns had greater
concentrations of metals (Tables S6 and S7) which interacted with P to form primarily Ca- and
Mg-$P_i$ species (Fig. 3; Table S2).






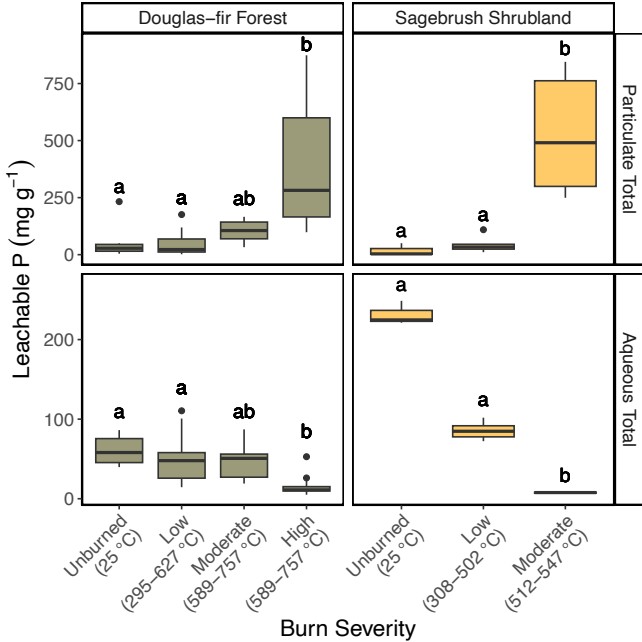

Figure 4. Relationship of burn severity and vegetation type with leachable P concentration (mg P g$^{-1}$; calculated from Equation 1) for total P in the particulate phase and total P in the aqueous phase. Molybdate-reactive P in the aqueous phase are reported in the SI. Ranges of maximum temperatures (°C) reached within a respective burn severity category are reported in parentheses. Letters denote post hoc findings of burn severity significant differences within a vegetation type. Note difference scales of the y-axis for the particulate and aqueous phases. See Figure S6 for leachable aqueous molybdate reactive P results.


Additional changes to char composition, including organic P speciation and pH, also likely
contributed to decreased aqueous P mobilization with increased burning. We found a decrease in
non-molybdate reactive aqueous P, which is largely composed of organic P species (Condron et
al., 2015), with increasing burn severity (mixed effect model interaction term: $p < 0.001$, Fig. S6)
indicating less mobilization of organic P species with burning. The amount of mobilized P
compounds from char is also related to pH, where less P compounds are released at higher pH
(Silber et al., 2010; Zheng et al., 2013). We found aqueous P mobilization had an inverse
relationship with pH for both Douglas-fir forest ($p < 0.001$; $r^2 = 0.45$) and sagebrush shrubland
($p < 0.001$, $r^2 = 0.97$; Fig. S7). Overall, changing chemical composition of the charred material
decreases solubility and therefore reduces aqueous P mobilization into the environment
(Robinson et al., 2018; Uchimiya et al., 2015; Wu et al., 2023b; Xu et al., 2016).



The extent of decreased aqueous P mobilization was vegetation dependent (interaction term of
mixed effect model; $p < 0.001$; Fig. 4). However, because both vegetation types had similar
percentages of Ca-$P_i$ ($p = 0.18$; $r^2 = 0.15$), it indicates additional controls on aqueous P
mobilization. In addition to Ca-$P_i$ and Mg-$P_i$, moderate-severity Douglas-fir forest contained P
compounds associated with Na (XANES: $22.7 \pm 22.1\%$) and organic P species (XANES: $6.9 \pm$
$11.9\%$; NMR: $4.4 \pm 4.2\%$), whereas sagebrush shrubland was <1% organic P (XANES: $0 \pm 0\%$;
NMR: $0.7 \pm 0.4\%$; Fig.s 2 and 3; Tables S2 and S5). Greater solubility of these chemical species
likely contributes to Douglas-fir forest moderate-severity burns mobilizing 6.4 times more
aqueous P than sagebrush shrubland ($p = 0.004$). Changing chemical speciation from soluble
organic P to less soluble inorganic species (Mukherjee and Zimmerman, 2013; Xu et al., 2016)
resulted in the decreased export of P compounds with increased burn severity and contributed to
the amount of P compounds mobilized from the respective vegetation types. This has important
implications for P compounds are transported in the environment because organic P can leach
faster than inorganic (McDowell et al., 2021).

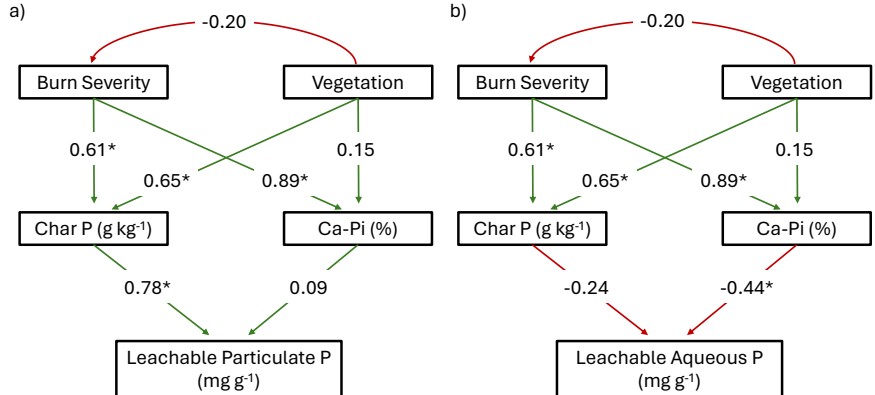

Figure 5. Path analysis model results for the impact of burn severity and vegetation type
on leachable P in the (a) particulate and (b) aqueous phases, as mediated by solid unburn
and char P concentration and chemical composition. All relationships are reported with
significance ($\alpha = 0.05$) denoted with an asterisk symbol on the standardized correlation
coefficient (analogous to relative regression weights). Paths are green for positive
relationships and red for negative. Leachable Particulate P Model: $\chi^2 = 16.277$, $p < 0.05$,
df = 3, RMSEA = 0.483, AIC = 108.3; Leachable Aqueous P Model: $\chi^2 = 19.032$, $p <$
0.05, df = 3, RMSEA = 0.530, AIC = 122.1. See Fig S4 for original hypothesized model.


## 4 Conclusions

Our objective was to understand how P concentration and composition of charred vegetation
changes along burned gradients to then influence the amount of P potentially mobilized into the
environment. We found systematic changes in P chemistry across vegetation types and
summarize these findings into a conceptual model (Fig. 6). Identifiable structures decreased with
increasing black charring and/or white ash with increasing burn severity. Total Ca, Fe, Al, K,
Ma, and Na concentrations increased while total C was lost as charring occurred. Solid char



concentration and composition controlled how P compounds were mobilized from burned
material. Overall, burning resulted in an increase of char P concentration, which subsequently
controlled the mobilization of particulate-bound P compounds from the chars. As burning
progressed, chars compositionally transitioned from proportionally more organic P species,
including both monoester and diesters, to Ca- and Mg- bound inorganic P species. These
compositional changes resulted in less soluble inorganic P species and therefore reduced aqueous
P mobilization in higher severity burns. Across vegetation types, chars became more divergent
from the unburned vegetation material in P composition and mobilization potential as burning
continued. Burn severity and vegetation type indirectly influenced the quantity and leachable
phase (i.e., particulate or aqueous) of P compounds that were mobilized from charred material by
altering solid sample concentration and composition.
Although both vegetation types followed similar concentration and compositional patterns,
sagebrush shrubland tended to appear 'more burned' than Douglas-fir forest in our P burn
severity conceptual model (Fig. 6). The P concentration of Douglas-fir forest chars and leachates
were more resilient to change with burning compared to sagebrush shrubland. For example,
generally, P transformations in sagebrush shrubland moderate-severity burns chemically
resembled that of Douglas-fir forest high-severity burns. Taken together, this indicates that
although sagebrush shrubland experiences more low- and moderate-severity burns (Stavi, 2019),
the response of P chemistry post-fire may resemble Douglas-fir forests burned at higher
severities. This response is important to note as shifts in fire severity are not occurring uniformly
across all ecosystem type (Francis et al., 2023; Halofsky et al., 2020; Reilly et al., 2017).

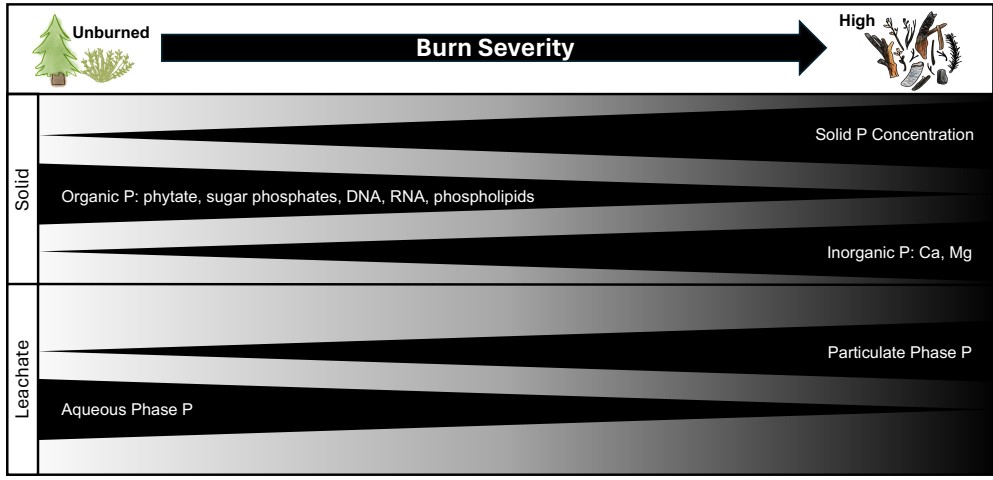

Figure 6. Conceptual framework for phosphorus biogeochemical shifts with increasing burn severity
where solid P concentration increases, organic P species decrease while inorganic P increases.
Leachates from the solid samples increased in mobilization of P in the particulate phase but
decreased in aqueous P with burning.




Organic soil horizons immobilize large amounts of P (Kruse et al., 2015) that can then serve as
an important source of bioavailable P after burning (Schaller et al., 2015). The key to
bioavailable P is that it can enter solution for subsequent uptake by plants and microbes (Kruse et
al., 2015). Our leaching experiments provide insight to the potential mobilization, and therefore
bioavailability, of P from solid vegetation chars. We found burning resulted in less P released
into the environment in the aqueous phase, although particulate-bound P increased and may be
an important source of available over longer timeframes. Our study helps to provide additional
information on the potential environmental fate of P post-fire in the context of different burn
severities and ecosystem types.

## Acknowledgments

We thank Christopher Myers for assistance with the ICP analyses and Sophia McKever for help
with measuring molybdate reactive P. This research was supported by the U.S. Department of
Energy, Office of Science, Office of Biological and Environmental Research, Environmental
System Science (ESS) Program. This contribution originates from the River Corridor Scientific
Focus Area project at Pacific Northwest National Laboratory (PNNL). PNNL is operated by
Battelle Memorial Institute for the United States Department of Energy under contract no. DE-
AC05-76RL01830. Portions of this research were performed on a project award
(10.46936/lser.proj.2021.51840/60000342) from the Environmental Molecular Science
Laboratory (EMSL) (grid.436923.9), a DOE Office of Science User Facility sponsored by the
Biological and Environmental Research program under Contract No. DE-AC05-76RL01830.
XANES data were collected from the Stanford Synchrotron Radiation Lightsource, SLAC
National Accelerator Laboratory, which is supported by the U.S. Department of Energy, Office
of Science, Office of Basic Energy Sciences under Contract No. DE-AC02-76SF00515. We
would like to give a special thanks to Erik Nelson, the beamline scientist from SSRL that helped
us collect those data.

## Code/Data availability

All data and code are publicly available on the Environmental System Science Data
Infrastructure for a Virtual Ecosystem (ESS-DIVE) repository (Grieger et al., 2022) or GitHub
(https://github.com/river-corridors-sfa/rcsfa-RC3-BSLE_P).

## Competing interests

The authors declare no competing interests.

## Author Contribution

Conceptualization: M.E.B., A.N.M.-P., J.A.R., K.D.B., E.B.G., S.G., T.D.S. Methodology and
Software: A.N.M.-P., M.E.B., S.G., J.D.B, K.B., E.B.G, P.A, V.A.G-C, K.M, J.A.R, R.P.Y.,
P.A.O., L.R. Investigation: M.E.B., P.A, S.G., K.M, L.R., J.A.R., J.D.B., K.D.B., R.P.Y. Data
Curation: M.E.B., S.G., P.A, V.A.G-C., A.N.M.-P., K.M, J.D.B., K.D.B., L.R., J.A.R. Formal
Analysis: M.E.B., A.N.M-P. J.A.R., V.A.G.-C., P.A., P.A.O., R.P.Y. Validation: M.E.B., P.A.,
K.M., V.A.G.-C., A.N.M-P., P.A.O., R.P.Y. Visualization: M.E.B., A.N.M.-P., S.G. Writing -



Original Draft: M.E.B., A.N.M.-P., J.A.R., S.G., K.M., R.P.Y. Writing – Review and Editing:
M.E.B., P.A., J.D.B., K.D.B., V.A.G.-C., E.B.G., A.N.M.-P., P.A.O., L.R., J.A.R, T.D.S., R.P.Y.

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
