# Peer review of "Burn severity and vegetation type control phosphorus"

_EGUsphere, 2025_

## Author Comment (AC1)

**Please find our responses given in bold blue text below after each individual comment or section provided by the reviewer.**

Brief Summary of the manuscript:

Barnes and co-authors investigate the effects of wildfire burn severity on phosphorus (P) mobilization in Douglas-fir forests and sagebrush shrublands. Through laboratory leaching experiments, they examine how different burn severities influence particulate and aqueous P release. The authors find that higher burn severity increases particulate P mobilization while decreasing aqueous P availability, with particulate P controlled by total char P and aqueous P driven by solubility changes. Using nuclear magnetic resonance and X-ray absorption spectroscopy, they show that organic P compounds are thermally mineralized into inorganic calcium- and magnesium-bound forms. The study highlights that fire severity and vegetation type drive post-fire shifts in P cycling, with implications for nutrient transport and ecosystem recovery.

The paper is well-written and presents compelling results on phosphorus (P) transformations following fire. The findings provide valuable insights into how burn severity and vegetation type influence P retention and mobilization. However, several issues need to be addressed before the manuscript is suitable for publication.

I recommend the paper for publication after moderate revision, focusing on clarifying key mechanisms, improving data presentation, and addressing inconsistencies in comparisons.

**Response: Thank you for the favorable review of our manuscript. In a revised manuscript, we will focus on clarifying the key mechanisms in the discussion section, improve the data presentation by including figures and tables in the main text that were previously only in the SI, and clarify text related to inconsistencies in comparisons in the discussion section.**

**Major Issues Requiring Revision:**

1. Unjustified Burn Severity Comparisons:

The authors compare moderate-severity burns in sagebrush shrublands to high-severity burns in Douglas-fir forests without justification. This prevents direct comparisons and raises concerns about bias in data interpretation. Either compare the same burn severities across vegetation types or provide a clear justification for the chosen comparison.

**Response: We will include more complete justification for the comparison between Douglas-fir and sagebrush at different severities done in the discussion section. We will do this by emphasizing that these are compared in the manuscript because they appear chemically similar.**

1.  Unclear Mechanisms of P Transformation and Mobilization:

The study claims burn severity influences P transformations, yet the chemical mechanisms behind these changes remain vague. For example, the authors state that aqueous P mobilization is "composition-controlled" by Ca-Pi, but later indicate that Ca-Pi concentrations are similar across vegetation types, suggesting other factors must be involved.

**Response: We will clarify language regarding P transformations, particularly when compared across the burn gradients or between vegetation types. We will also revise the discussion around path analysis design and results, by clarifying lines 264-266 and 444-481, and in the Conclusions 493-497. Namely, we included only Ca-Pi in the path analysis to aid in interpretation of the path analysis results, but in reality, there are other drivers of aqueous P mobilization such as Mg-Pi, organic P speciation, and pH. We will revise other locations in the manuscript that suggests Ca-Pi as the only driver of aqueous P mobilization.**

1.  Fire Temperatures in Experimental Burns Are Lower Than Real Wildfires:

The highest recorded burn temperature for sagebrush (530°C) and Douglas-fir (704°C) is significantly lower than real wildfire conditions, which can exceed 1,000°C. Since P volatilization occurs above ~700°C, the study may underestimate P losses in real wildfire conditions. Discuss how P retention might differ if sagebrush shrubland was burned at higher temperatures (e.g., 800–1,200°C).

**Response: We will add in additional context about how burn temperature and duration in our experiments relate to natural wildfires. We note that the aim of using our experimental burns was to better represent field burning conditions than what is currently commonly used in the P chemistry community; most of the literature on chemistry post-wildfire are based on burning materials in ovens, which is not representative of heterogenous burns that are common of field conditions (Brucker et al., 2022). Temperature is included in our study to better compare back to oven-based studies. Therefore, although we may not reach maximum temperatures experienced by a natural wildfire in our experiments, we believe the range of temperatures and durations of heating achieved by our experimental burns are largely representative of a large range in temperature and durations of heating experienced during wildfire (as noted in a**

similar study Brucker et al., 2024). We will reference these studies and this justification in the revised manuscript.

References:

Brucker, Carli P., et al. "A review of simulation experiment techniques used to analyze wildfire effects on water quality and supply." *Environmental Science: Processes & Impacts* 24.8 (2022): 1110-1132.

Brucker, Carli P., et al. "A laboratory-scale simulation framework for analysing wildfire hydrologic and water quality effects." *International Journal of Wildland Fire* 33.12 (2024).

1. Post-Fire Ecosystem Recovery:

The manuscript discusses P mobilization and transformation but does not address how these changes affect ecosystem recovery after fire. It is unclear whether particulate P will eventually become bioavailable or remain locked in ash.

Response: We will modify lines 513-521 to include more discussion on how our findings on P concentration, composition, and transport from burned material alter bioavailability by revisiting concepts from the Introduction (lines 93-98). We will also discuss how our results may translate to ecosystem recovery across different timescales using key examples in the literature (i.e., Santin et al., 2018, Silins et al., 2014, Emelko et al., 2016, Bodi et al., 2014, and Rust et al., 2018).

References:

Santín, C., Otero, X. L., Doerr, S. H., and Chafer, C. J.: Impact of a moderate/high-severity prescribed eucalypt forest fire on soil phosphorous stocks and partitioning, Sci. Total Environ., 621, 1103–1114, 2018.

Silins, U., Bladon, K. D., Kelly, E. N., Esch, E., Spence, J. R., Stone, M., Emelko, M. B., Boon, S., Wagner, M. J., Williams, C. H. S., and Tichkowsky, I.: Five-year legacy of wildfire and salvage logging impacts on nutrient runoff and aquatic plant, invertebrate, and fish productivity, Ecohydrol., 7, 1508–1523, 2014.

Emelko, Monica B., et al. "Sediment-phosphorus dynamics can shift aquatic ecology and cause downstream legacy effects after wildfire in large river systems." *Global Change Biology* 22.3 (2016): 1168-1184.

Bodí, M. B., Martin, D. A., Balfour, V. N., Santín, C., Doerr, S. H., Pereira, P., Cerdà, A., and Mataix-Solera, J.: Wildland fire ash: Production, composition and eco-hydro-geomorphic effects, Earth-Sci. Rev., 130, 103–127, 2014.

Rust, A. J., Hogue, T. S., Saxe, S., and McCray, J.: Post-fire water-quality response in the western United States, Int. J. Wildland Fire, 27, https://doi.org/10.1071/WF17115, 2018.

Missing Data or Discussion on Ash Color:

The conclusion states that ash color (black charring and white ash) increases with burn severity, yet this was not discussed in the results or presented in any figure or table. If ash color was recorded, include data or observations in the results and discuss its significance. If it was not recorded, remove the statement from the conclusion.

**Response: We will revise this part of the conclusions (see responses below in minor revisions) and refer to Figure S4. We will remove word "appear" in the referenced conclusion, which decreases clarity in the intended meaning.**

Minor Revisions:

Abstract:

Lines 33–34: The magnitude of P mobilization (e.g., "Burning increased particulate P mobilization (6.9-fold and 29-fold) but decreased aqueous P release (3.8-fold and 30.5-fold)") varies significantly between Douglas-fir and sagebrush. Why? Briefly mention the mechanism driving these differences.

**Response: We will revise this sentence and the following on in the abstract for clarity. We will add in an additional sentence bridging between the two sentences that will state the following "Particulate and dissolved phases were under contrasting mechanisms."**

Line 29–31: The sentence "However, it is unclear if post-fire responses are primarily driven by changes to the molecular composition of the charred material or from the transport of P-containing compounds." is difficult to follow. Consider rewording.

**Response: We will reword the sentence as follows "However, it is unclear if shifts in P composition or P concentration are responsible for changes in P dynamics post-fire".**

Line 39: "Thermally mineralized to inorganic P moieties"—clarify how this affects P availability in soils.

**Response: After this sentence we will add in a short description of how this impacts P availability/solubility to the end of the sentence.**

Introduction:

Lines 60–63 and 67-68: Long sentences—consider breaking them up for clarity.

**Response: As suggested by the reviewer, we will break up the sentences on lines 60-63 and 67-68 to increase clarity.**

Lines 111–119: Burn severity should be introduced earlier when discussing fire intensity and nutrient cycling.

**Response: We will introduce burn severity on line 67, when we first discuss these concepts.**

Methods:

What was the collection timeframe? Seasonal variations can influence plant moisture content, affecting fire behavior and P release.

**Response: We will add in additional details to clarify the study design. Exact conditions are detailed in our data package, Grieger et al., 2022 which we will also reference here. Please note that we did manipulate moisture conditions of the vegetation prior to burning as part of our experimental design.**

**References:**
**Grieger, S., Bailey, J., Barnes, M., Bladon, K. D., Forbes, B., Garayburu-Caruso, V. A., Graham, E. B., Goldman, A. E., Homolka, K., McKever, S. A., Myers-Pigg, A., Otenburg, O., Renteria, L., Roebuck, A., Scheibe, T. D., and Torgeson, J. M.: Organic Matter Concentration and Composition of Experimentally Burned Open Air and Muffle Furnace Vegetation Chars across Differing Burn Severity and Feedstock Types from Pacific Northwest, USA (V3), https://doi.org/10.15485/1894135., 2022.**

The geographic description ("Pacific Northwest, USA") is too vague. Include specific sites or coordinates for clarity.

**Response: We will add in additional details to clarify the study design. Briefly, vegetation was collected to be representative of different land cover types present across the Pacific Northwest and we are not representing exact sample sites or locations, but rather archetypic dominant vegetation across PNW ecosystems. See Figure 1 in Roebuck et al. 2025 and description of vegetation and burn experiments in Myers-Pigg et al., 2024. If of interest, exact sites where the representative vegetation was collected can be found in our data package, Grieger et al., 2022. We will add references to these other studies in the main text in this section, to clarify the study design.**

**References:**

Grieger, S., Bailey, J., Barnes, M., Bladon, K. D., Forbes, B., Garayburu-Caruso, V. A., Graham, E. B., Goldman, A. E., Homolka, K., McKever, S. A., Myers-Pigg, A., Otenburg, O., Renteria, L., Roebuck, A., Scheibe, T. D., and Torgeson, J. M.: Organic Matter Concentration and Composition of Experimentally Burned Open Air and Muffle Furnace Vegetation Chars across Differing Burn Severity and Feedstock Types from Pacific Northwest, USA (V3), https://doi.org/10.15485/1894135., 2022.

Myers-Pigg, A. N., Grieger, S., Roebuck, J. A., Jr, Barnes, M. E., Bladon, K. D., Bailey, J. D., Barton, R., Chu, R. K., Graham, E. B., Homolka, K. K., Kew, W., Lipton, A. S., Scheibe, T., Toyoda, J. G., and Wagner, S.: Experimental Open Air Burning of Vegetation Enhances Organic Matter Chemical Heterogeneity Compared to Laboratory Burns, Environ. Sci. Technol., 58, 9679–9688, 2024.

Roebuck, J. A., Jr, Grieger, S., Barnes, M. E., Gillespie, X., Bladon, K. D., Bailey, J. D., Graham, E. B., Chu, R., Kew, W., Scheibe, T. D., and Myers-Pigg, A. N.: Molecular shifts in dissolved organic matter along a burn severity continuum for common land cover types in the Pacific Northwest, USA, Sci. Total Environ., 958, 178040, 2024.

Lines 138–140: The statement that "Douglas-fir burns at higher intensities due to fuel loading, while sagebrush burns at lower intensities" is too general. Explain why these fuel differences affect fire behavior.

**Response: Thank you for bringing this to our attention. We plan to revise this section to increase clarity in our meaning here. Briefly, we will 1) add in several sentences about the design and rationale for the vegetation studied from our data package Grieger et al., 2022 and 2) bring in relevant literature on burn severity differences observed across fires in Douglas-fir and sagebrush ecosystems (Stavi, 2019; Roebuck et al., 2024 Fig 1; Halofsky et al., 2020; Reilly et al., 2017).**

**References:**

**Grieger, S., Bailey, J., Barnes, M., Bladon, K. D., Forbes, B., Garayburu-Caruso, V. A., Graham, E. B., Goldman, A. E., Homolka, K., McKever, S. A., Myers-Pigg, A., Otenburg, O., Renteria, L., Roebuck, A., Scheibe, T. D., and Torgeson, J. M.: Organic Matter Concentration and Composition of Experimentally Burned Open Air and Muffle Furnace Vegetation Chars across Differing Burn Severity and Feedstock Types from Pacific Northwest, USA (V3), https://doi.org/10.15485/1894135., 2022.**

**Roebuck, J. A., Jr, Grieger, S., Barnes, M. E., Gillespie, X., Bladon, K. D., Bailey, J. D., Graham, E. B., Chu, R., Kew, W., Scheibe, T. D., and Myers-Pigg, A. N.: Molecular**

shifts in dissolved organic matter along a burn severity continuum for common land cover types in the Pacific Northwest, USA, Sci. Total Environ., 958, 178040, 2024.

Halofsky, Jessica E., David L. Peterson, and Brian J. Harvey. "Changing wildfire, changing forests: the effects of climate change on fire regimes and vegetation in the Pacific Northwest, USA." *Fire Ecology* 16.1 (2020): 1-26.

Reilly, Matthew J., et al. "Contemporary patterns of fire extent and severity in forests of the Pacific Northwest, USA (1985–2010)." *Ecosphere* 8.3 (2017): e01695.

Lines 146–147: What ratio of woody to canopy material was used? The relative proportion of wood vs. foliage affects combustion properties and nutrient release.

Response: These details exist in our accompanying data package, Grieger et al., 2022. We will include relevant details in the revised manuscript text directly. Briefly, the ratio was 40% canopy (materials <0.5cm) and 60% woody (materials >0.5cm). We will also reference the data package in this part of the manuscript.

References:

Grieger, S., Bailey, J., Barnes, M., Bladon, K. D., Forbes, B., Garayburu-Caruso, V. A., Graham, E. B., Goldman, A. E., Homolka, K., McKever, S. A., Myers-Pigg, A., Otenburg, O., Renteria, L., Roebuck, A., Scheibe, T. D., and Torgeson, J. M.: Organic Matter Concentration and Composition of Experimentally Burned Open Air and Muffle Furnace Vegetation Chars across Differing Burn Severity and Feedstock Types from Pacific Northwest, USA (V3), https://doi.org/10.15485/1894135., 2022.

No mention of initial sample preparation—were plant materials cleaned, dried, or processed before burning?

Response: These details exist in our accompanying data package, Grieger et al., 2022. We will include relevant details in the revised manuscript. Briefly, all plant materials were air dried for at least two weeks prior to burning. We also manipulated fuel moisture before burn experiments and kept living and dead plant materials separate. We will also reference the data package in this part of the manuscript.

References:

Grieger, S., Bailey, J., Barnes, M., Bladon, K. D., Forbes, B., Garayburu-Caruso, V. A., Graham, E. B., Goldman, A. E., Homolka, K., McKever, S. A., Myers-Pigg, A., Otenburg, O., Renteria, L., Roebuck, A., Scheibe, T. D., and Torgeson, J. M.:

**Organic Matter Concentration and Composition of Experimentally Burned Open Air and Muffle Furnace Vegetation Chars across Differing Burn Severity and Feedstock Types from Pacific Northwest, USA (V3), https://doi.org/10.15485/1894135., 2022.**

Results and Discussion:

Lines 282–284: The study may underestimate P volatilization since wildfires can exceed 1,000°C, causing greater P losses. Acknowledge this limitation.

**Response: We will add in acknowledgement of these limitations, in the context of our experimental burning results in relation to a real fire. In particular, we will emphasize the heterogeneity of burning that the open air simulations experienced, which are representative of heterogenous wildfire burning conditions.**

Lines 292–302: The comparison between moderate-severity sagebrush and high-severity Douglas-fir needs clear justification.

**Response: We will clarify language throughout the manuscript where direct comparisons between Douglas-fir and sagebrush at different severities are being made and remove comparisons that are unclear.**

Lines 308–313: Which forms of P are retained vs. combusted? Clarify whether organic or inorganic P compounds are responsible.

**Response: We are specifically focused on the conversion of organic P to inorganic P in this section and will revise the text accordingly.**

Lines 328–331: The phrase "selective protection" explain what protects P from mineralization.

**Response: This phrase was in reference to physical protection of P, such as mineral aggregates. We will clarify this in the text.**

Lines 373–395: The manuscript claims pyrophosphate forms from orthophosphate, but why did sagebrush chars produce less pyrophosphate than Douglas-fir chars, despite having more phytate initially?

**Response: Pyrophosphate can be produced either from orthophosphate or phytate. We will clarify the language around the discussion of the two pathways that pyrophosphate is produced from in the manuscript text.**

Lines 460–463: The mechanism by which pH affects P solubility is not clearly explained. Report actual pH values measured.

**Response: Thank you for this comment, we will bring up the pH figure from the Supplemental materials into the main text.**

Lines 478–479: "This has important implications for P compounds are transported…" grammatical error—revise for clarity.

**Response: We will correct this to say "This has important implications for P compounds that are transported…"**

Conclusions:

Lines 483–485: The first sentence should summarize the main findings upfront before interpretation.

**Response: We will add in a high-level summary of our main findings at the start of the conclusions section.**

Lines 486–487: The conceptual model (Fig. 6) is mentioned but not explained. Briefly describe its significance.

**Response: The conceptual model was intended to synthesize our main findings. We will increase discussion of the conceptual model in our conclusions section.**

Lines 500–503: The phrase "more burned" is vague—does this mean greater P retention, more mass loss, or another factor?

**Response: Here we meant greater P transformations and will revise the text to reflect this.**

Lines 506–509: The statement on shifting fire severity lacks context—explain why this matters for P retention and post-fire recovery.

**Response: This is a great point. We will be more explicit about the P changes that occur with severity that we discovered in our study and how that relates to P in the ecosystem post-fire.**

Lines 510–517: Organic soil horizons are introduced but were not a focus of the study. Instead, discuss how P transformations affect bioavailability over time.

**Response:** The P studied here are the starting (litter) materials that are available for subsequent overland transport and do not include soils, and we agree mention of organic soils is confusing in this context. Therefore, we will modify this discussion to focus on the potential for different P pools (organic/inorganic, particulate/dissolved) to have different cycling in the environment (e.g. dissolved phase will be more reactive/have quicker cycling, solid P may be longer term source of P to the environment) compared to starting vegetation P.

---

## Author Comment (AC2)

**Please find our responses given in bold blue text below after each individual comment or section provided by the reviewer.**

**Dear authors,**

This paper is an important contribution to understanding the biogeochemical impacts of fire on phosphorus fate and mobility. The study employs a robust methodological approach, including NMR and XANES analyses, to investigate how burn severity and vegetation type influence phosphorus composition and mobilization. The findings are relevant for understanding post-fire nutrient cycling and have implications for both terrestrial and aquatic ecosystems.

**Response: Thank you for your favorable review of our manuscript.**

**Major Comments:**

1. Providing information on the geological and soil characteristics of the study site, including soil type and pH, would help contextualize phosphorus dynamics. Additionally, details on climate conditions such as rainfall, temperature, and seasonality would improve the interpretation of post-fire phosphorus mobilization and retention. If the bedrock is limestone, it could increase soil calcium concentrations and pH, reducing bioavailable phosphorus and promoting the formation of Ca-bound phosphorus (Ca-P). This may influence the phosphorus speciation in the original soil, affecting the composition of the phosphorus pool from which the organic samples were taken. If available, correlating soil properties with phosphorus fractions could further strengthen the study.

    **Response: This study was designed to examine changes across land cover archetypes across the Pacific Northwest, not a specific site study. This is notably a laboratory experiment and simulates P mobilization by leaching chars with artificial rainwater and does not include study of soils or in-situ leachates from litter/overland flow. We will clarify these points throughout the introduction and method section to increase clarity on our study design and limitations.**

2. **Soil Phosphorus Pools and Pre-Fire Conditions:**

    It would be valuable to include data on the total phosphorus concentration in the soils from which the organic matter was sourced. Since the initial phosphorus pools in the soil can influence post-fire phosphorus mobilization and retention, this information could provide important context for interpreting the results. If this

data is available, correlating soil phosphorus status with the observed trends would further strengthen the study.

**Response: The organic matter burned here was plant material (i.e. litter), not soils. The initial pools of P in the study can be considered that of the unburned plant materials presented throughout the manuscript. We will clarify our description of the study design in response to this and the previous comment.**

3. **Use of the Term "Total P Concentration"** (Line 99 and elsewhere):

   The term "total P concentration" is unclear. It would be helpful to clarify whether this refers to the sum of all phosphorus fractions or a specific measurement of P concentration. Consistently defining this term throughout the text would improve clarity.

   **Response: Total P is derived from ICP-OES measurements. We will clearly state this in the methods and upon first usage.**

**Specific Comments:**

- **Line 65:** Expand on the impacts of climate change on fire regimes. The statement that fires are expected to increase in intensity and severity could be strengthened by elaborating on the mechanisms driving this trend, as climate change is expected to exacerbate fire risk.

  **Response: We agree with the reviewer there are several factors exacerbated by a changing climate that are influencing fire regimes in our study region. We will increase discussion of this in the introduction, and cite relevant literature such as Roebuck et al., 2025; Francis et al., 2023, Halofsky et al., 2020 and Reilly et al., 2017.**

  **References:**
  **Roebuck Jr, J. Alan, et al. "Molecular shifts in dissolved organic matter along a burn severity continuum for common land cover types in the Pacific Northwest, USA."** *Science of the Total Environment* **958 (2025): 178040.**
  **Francis, Emily J., et al. "Proportion of forest area burned at high-severity increases with increasing forest cover and connectivity in western US watersheds."** *Landscape Ecology* **38.10 (2023): 2501-2518.**
  **Halofsky, Jessica E., David L. Peterson, and Brian J. Harvey. "Changing wildfire, changing forests: the effects of climate change on fire regimes**

and vegetation in the Pacific Northwest, USA." *Fire Ecology* **16.1 (2020): 1-26.**

**Reilly, Matthew J., et al. "Contemporary patterns of fire extent and severity in forests of the Pacific Northwest, USA (1985–2010)."** *Ecosphere* **8.3 (2017): e01695.**

- **Line 285 (Figure 1):** The distinction between moderate and high temperatures for Douglas fir is unclear. The figure appears to show overlapping temperature ranges - maybe you can think of a different phrasing? Additionally, the lines on the figure should be explained—do they represent the 25th and 75th percentiles? Are they median or mean values?

  **Response: Thank you for bringing this point of confusion to our attention. Burn severity was assessed via visual metrics, which includes ash color, degree of charring, and degree of consumption (Parsons et al., 2010). Therefore, burn severity is not just dependent on temperature, but is also impacted by duration of heating and additional fuel loading metrics. We had originally included temperature in our figure axes to increase comparability with temperature-based studies of P chemical changes common in the literature. We will move Table S4 to the main text to better set up our burn severity definition and details of the burn table experiments studied herein. We will also point to the SI figure on burn severity (Figure S1) in the main text.**

  **Regarding Figure 1, this is a boxplot. We will annotate in the figure legend this detail to reduce possible confusion on interpretation, remove the temperature ranges on the axis, and remove colors from the boxplots for simplicity. We will refer to the table brought into the main text from the SI for sample numbers of each observation and the temperature ranges for each severity.**

  **References:**

  **Parson, Annette, et al. "Field guide for mapping post-fire soil burn severity."** *Gen. Tech. Rep. RMRS-GTR-243. Fort Collins, CO: US Department of Agriculture, Forest Service, Rocky Mountain Research Station. 49 p.* **243 (2010).**

- **Line 334 (Figure 2):** The color scheme should be adjusted to better differentiate between organic P and inorganic P. Additionally, the use of "moderate" and "high" temperatures for Douglas-fir is unclear, as it appears inconsistent with figure 1.

**Response: We will add in hashing onto the colors for organic P in both this figure and figure 3 to increase clarity and remove the temperatures from figure legend.**

- **Line 453 (Figure 3):** Similar to Figure 1, further clarification is needed. What do the dots above the boxes represent? How many samples were measured? The figure suggests that Douglas-fir moderate fire has significantly higher leachable P than low-temperature burns, and that high-temperature burns are significantly higher than moderate burns. For sagebrush, in the lower right panel, the "a" and "a" labels above the boxes indicate significant differences, yet the values appear different. Please review and clarify.

  **Response: We believe this comment is in reference to figure 4, not figure 3. Similar to figure 1, we will clearly state in the figure legend that this is a boxplot. We will also remove the temperature ranges on the axis and remove colors from the boxplots for simplicity. We will also clarify the lettering for significant differences in the figure caption (briefly, those with the same lettering above them are not significantly different). We will refer to the table brought into the main text from the SI for sample numbers of each observation and the temperature ranges for each severity.**

- **Line 455:** The impact of pH on phosphorus solubility is mentioned, but actual pH values are not provided. I suggest including these values in the results.

  **Response: In response to this comment and a similar comment from R1, we will move Figure S7 from the SI into the main text, which shows our pH data.**

**Line 470:** The proportion of Na-P is relatively high in Douglas-fir. A brief discussion of the potential role of Na-P in post-fire phosphorus dynamics would be useful.

**Response: Our current understanding of P biogeochemistry is largely around Ca-, Al-, and Fe-P given the popularity of sequential chemical fractionation schemes, which infer speciation of these compounds based on solubility (Kruse et al 2015). XANES is increasingly being used to identify P speciation within environmental samples, but many studies still focus on Ca-, Al- and Fe-P compound identification. However, other studies characterizing chars have identified Na-P (i.e., Rose et al 2019) and Li and Brett (2013) have identified sodium tripolyphosphate as having high nutrient uptake and bioavailability during bioassay experiments. We will add these emerging viewpoints into our discussion section.**

**References:**

Kruse, Jens, et al. "Innovative methods in soil phosphorus research: A review." *Journal of plant nutrition and soil science* 178.1 (2015): 43-88.

Rose, Terry J., et al. "Phosphorus speciation and bioavailability in diverse biochars." *Plant and Soil* 443 (2019): 233-244.

Li, Bo, and Michael T. Brett. "The influence of dissolved phosphorus molecular form on recalcitrance and bioavailability." *Environmental Pollution* 182 (2013): 37-44.

---

## Author Response (AR1)

**Please find our responses given in bold blue text below after each individual comment or section provided by the reviewers. All line numbers are in reference to the clean, non-track changed document.**

**Reviewer 1:**

Brief Summary of the manuscript:

Barnes and co-authors investigate the effects of wildfire burn severity on phosphorus (P) mobilization in Douglas-fir forests and sagebrush shrublands. Through laboratory leaching experiments, they examine how different burn severities influence particulate and aqueous P release. The authors find that higher burn severity increases particulate P mobilization while decreasing aqueous P availability, with particulate P controlled by total char P and aqueous P driven by solubility changes. Using nuclear magnetic resonance and X-ray absorption spectroscopy, they show that organic P compounds are thermally mineralized into inorganic calcium- and magnesium-bound forms. The study highlights that fire severity and vegetation type drive post-fire shifts in P cycling, with implications for nutrient transport and ecosystem recovery.

The paper is well-written and presents compelling results on phosphorus (P) transformations following fire. The findings provide valuable insights into how burn severity and vegetation type influence P retention and mobilization. However, several issues need to be addressed before the manuscript is suitable for publication.

I recommend the paper for publication after moderate revision, focusing on clarifying key mechanisms, improving data presentation, and addressing inconsistencies in comparisons.

**Response: Thank you for the favorable review of our manuscript. We have focused on clarifying key mechanisms in the discussion section, improved the data presentation by including figures and tables in the main text that were previously only in the SI, and clarified the text related to inconsistencies in comparisons in the discussion section.**

**Major Issues Requiring Revision:**

1. Unjustified Burn Severity Comparisons:

The authors compare moderate-severity burns in sagebrush shrublands to high-severity burns in Douglas-fir forests without justification. This prevents direct comparisons and raises concerns about bias in data interpretation. Either compare the

same burn severities across vegetation types or provide a clear justification for the chosen comparison.

**Response: We have included justification for why these comparisons were made upon first comparison (i.e. they were the highest burn severity reached by either treatment) in the results and discussion section lines 311-313.**

1. Unclear Mechanisms of P Transformation and Mobilization:

The study claims burn severity influences P transformations, yet the chemical mechanisms behind these changes remain vague. For example, the authors state that aqueous P mobilization is "composition-controlled" by Ca-Pi, but later indicate that Ca-Pi concentrations are similar across vegetation types, suggesting other factors must be involved.

**Response: We have clarified language regarding P transformations. We have revised the discussion around path analysis design and results on lines 466-471. Namely, we included only Ca-Pi in the path analysis to aid in interpretation of the path analysis results, but in reality, there are other drivers of aqueous P mobilization such as Mg-Pi, organic P speciation, and pH.**

1. Fire Temperatures in Experimental Burns Are Lower Than Real Wildfires:

The highest recorded burn temperature for sagebrush (530°C) and Douglas-fir (704°C) is significantly lower than real wildfire conditions, which can exceed 1,000°C. Since P volatilization occurs above ~700°C, the study may underestimate P losses in real wildfire conditions. Discuss how P retention might differ if sagebrush shrubland was burned at higher temperatures (e.g., 800–1,200°C).

**Response: We have added additional context about how burn temperature and duration in our experiments relate to natural wildfires, on lines 286-290. We note that the aim of using our experimental burns was to better represent field burning conditions than what is currently commonly used in the P chemistry community; most of the literature on chemistry post-wildfire are based on burning materials in ovens, which is not representative of heterogenous burns that are common of field conditions (Brucker et al., 2022). Therefore, although we may not reach maximum temperatures experienced by a natural wildfire in our experiments, we believe the range of temperatures and durations of heating achieved by our experimental burns are largely representative of a large range in temperature and durations of heating experienced during wildfire (as noted in a similar study Brucker et al., 2024).**

**References:**

Brucker, Carli P., et al. "A review of simulation experiment techniques used to analyze wildfire effects on water quality and supply." *Environmental Science: Processes & Impacts* 24.8 (2022): 1110-1132.

Brucker, Carli P., et al. "A laboratory-scale simulation framework for analysing wildfire hydrologic and water quality effects." *International Journal of Wildland Fire* 33.12 (2024).

1. Post-Fire Ecosystem Recovery:

The manuscript discusses P mobilization and transformation but does not address how these changes affect ecosystem recovery after fire. It is unclear whether particulate P will eventually become bioavailable or remain locked in ash.

**Response: We have included discussion on how our findings on P concentration, composition, and transport from burned material alter bioavailability by revisiting concepts from the introduction and discussed how our results may translate to ecosystem recovery across different timescales using key examples in the literature (i.e., Santin et al., 2018, Silins et al., 2014, Emelko et al., 2016, Bodi et al., 2014, and Rust et al., 2018), lines 538-551.**

**References:**

Santín, C., Otero, X. L., Doerr, S. H., and Chafer, C. J.: Impact of a moderate/high-severity prescribed eucalypt forest fire on soil phosphorous stocks and partitioning, Sci. Total Environ., 621, 1103–1114, 2018.

Silins, U., Bladon, K. D., Kelly, E. N., Esch, E., Spence, J. R., Stone, M., Emelko, M. B., Boon, S., Wagner, M. J., Williams, C. H. S., and Tichkowsky, I.: Five-year legacy of wildfire and salvage logging impacts on nutrient runoff and aquatic plant, invertebrate, and fish productivity, Ecohydrol., 7, 1508–1523, 2014.

Emelko, Monica B., et al. "Sediment-phosphorus dynamics can shift aquatic ecology and cause downstream legacy effects after wildfire in large river systems." *Global Change Biology* 22.3 (2016): 1168-1184.

Bodí, M. B., Martin, D. A., Balfour, V. N., Santín, C., Doerr, S. H., Pereira, P., Cerdà, A., and Mataix-Solera, J.: Wildland fire ash: Production, composition and eco-hydro-geomorphic effects, Earth-Sci. Rev., 130, 103–127, 2014.

Rust, A. J., Hogue, T. S., Saxe, S., and McCray, J.: Post-fire water-quality response in the western United States, Int. J. Wildland Fire, 27, https://doi.org/10.1071/WF17115, 2018.

Missing Data or Discussion on Ash Color:

The conclusion states that ash color (black charring and white ash) increases with burn severity, yet this was not discussed in the results or presented in any figure or table. If

ash color was recorded, include data or observations in the results and discuss its significance. If it was not recorded, remove the statement from the conclusion.

**Response: We have reworked this part of the conclusions section to increase clarity and referenced our supplemental figure. It now reads:**

**"From unburned to high-severity, identifiable structures decreased with increasing black charring and/or white ash with increasing burn severity (Fig. 7 panel 1; Fig. S1)."**

Minor Revisions:

Abstract:

Lines 33–34: The magnitude of P mobilization (e.g., "Burning increased particulate P mobilization (6.9-fold and 29-fold) but decreased aqueous P release (3.8-fold and 30.5-fold)") varies significantly between Douglas-fir and sagebrush. Why? Briefly mention the mechanism driving these differences.

**Response: The sentence "The mechanisms driving particulate and dissolved phase P compound mobilization were contrasting" was added on lines 34-35.**

Line 29–31: The sentence "However, it is unclear if post-fire responses are primarily driven by changes to the molecular composition of the charred material or from the transport of P-containing compounds." is difficult to follow. Consider rewording.

**Response: This sentence has been reworded to:**

**"However, it is unclear if shifts in P composition or P concentration are responsible for changes in P dynamics post-fire".**

Line 39: "Thermally mineralized to inorganic P moieties"—clarify how this affects P availability in soils.

**Response: We have added ", which will decrease P solubility" to the end of this sentence.**

Introduction:

Lines 60–63 and 67-68: Long sentences—consider breaking them up for clarity.

**Response: These sentences have been edited for clarity. They now read:**

**"Organic and inorganic nutrient pools and fluxes can be altered by burning through multiple mechanisms. These include the loss of volatile compounds, altered physiochemical properties from the incomplete combustion of organic material (from partially charred biomass to ash; collectively referred to as chars (Bird et al., 2015)), and enhanced material transport from leaching and erosion (Bodí et al., 2014)."**

**And**

**"Fire frequency, intensity, severity, and total area burned are expected to increase in many regions, such as the western United States (Doerr and Santín, 2016; Haugo et al., 2019; Jolly et al., 2015). In particular, in the Pacific Northwest, burn severity and area burn have increased in recent decades (Francis et al., 2023; Halofsky et al., 2020; Reilly et al., 2017; Roebuck et al., 2024)."**

Lines 111–119: Burn severity should be introduced earlier when discussing fire intensity and nutrient cycling.

**Response: We introduce burn severity upon first usage in the manuscript, now lines 66-67.**

Methods:

What was the collection timeframe? Seasonal variations can influence plant moisture content, affecting fire behavior and P release.

**Response: We have added in additional details on our study design throughout Section 2.1.**

The geographic description ("Pacific Northwest, USA") is too vague. Include specific sites or coordinates for clarity.

**Response: In response to this and the previous comment, we have edited our methods description Section 2.1.**

Lines 138–140: The statement that "Douglas-fir burns at higher intensities due to fuel loading, while sagebrush burns at lower intensities" is too general. Explain why these fuel differences affect fire behavior.

**Response: Thank you for bringing this to our attention. Section 2.1 has been edited throughout to increase clarity in response to this and other comments.**

Lines 146–147: What ratio of woody to canopy material was used? The relative proportion of wood vs. foliage affects combustion properties and nutrient release.

**Response: 40% to 60%; these details have been added into Section 2.1.**

No mention of initial sample preparation—were plant materials cleaned, dried, or processed before burning?

**Response: All plant materials were dried before burning. These details have been added into Section 2.1.**

Results and Discussion:

Lines 282–284: The study may underestimate P volatilization since wildfires can exceed 1,000°C, causing greater P losses. Acknowledge this limitation.

**Response: We have added acknowledgement of these limitations, in the context of our experimental burning results in relation to a real fire. The addition reads:**

**"In particular, while our burn treatments did not reach temperatures that would result in P volatilization, they did represent heterogenous burn conditions, incorporating a variety of burn durations and temperature ranges (Grieger et al., 2022; Myers-Pigg et al., 2024) that are consistent with other open air burn experiments (Brucker et al., 2022, 2024) (Table 1; Fig. S1)."**

Lines 292–302: The comparison between moderate-severity sagebrush and high-severity Douglas-fir needs clear justification.

**Response: We have clarified language throughout the manuscript where direct comparisons between Douglas-fir and sagebrush at different severities are being made and removed comparisons that are unclear. Example, lines 311-317.**

Lines 308–313: Which forms of P are retained vs. combusted? Clarify whether organic or inorganic P compounds are responsible.

**Response: We are specifically focused on the conversion of organic P to inorganic P in this section and have revised the text accordingly. Lines 327-328.**

Lines 328–331: The phrase "selective protection" explain what protects P from mineralization.

**Response: This phrase was in reference to physical protection of P, such as mineral aggregates and we have clarified this in the text lines 348-349.**

Lines 373–395: The manuscript claims pyrophosphate forms from orthophosphate, but why did sagebrush chars produce less pyrophosphate than Douglas-fir chars, despite having more phytate initially?

**Response: Pyrophosphate can be produced either from orthophosphate or phytate. We have modified the discussion around this point on lines 392-394.**

Lines 460–463: The mechanism by which pH affects P solubility is not clearly explained. Report actual pH values measured.

**Response: Thank you for this comment, we now include Figure 6 which shows our actual pH values.**

Lines 478–479: "This has important implications for P compounds are transported…" grammatical error—revise for clarity.

**Response: This has been corrected to say: "This has important implications for P compounds that are transported…"**

Conclusions:

Lines 483–485: The first sentence should summarize the main findings upfront before interpretation.

**Response: We have modified the start of our conclusions section to summarize the main findings of this work, lines 509-511.**

Lines 486–487: The conceptual model (Fig. 6) is mentioned but not explained. Briefly describe its significance.

**Response: The conceptual model was intended to synthesize our main findings. We have increased the discussion of the conceptual model throughout our Section 4.**

Lines 500–503: The phrase "more burned" is vague—does this mean greater P retention, more mass loss, or another factor?

**Response: Here we meant greater P transformations and have revised the text to reflect this, lines 526-528.**

Lines 506–509: The statement on shifting fire severity lacks context—explain why this matters for P retention and post-fire recovery.

**Response: This is a great point. We have modified this discussion on lines 532-538.**

Lines 510–517: Organic soil horizons are introduced but were not a focus of the study. Instead, discuss how P transformations affect bioavailability over time.

**Response: The P studied here are the starting (litter) materials that are available for subsequent overland transport and do not include soils, and we agree mention of organic soils is confusing in this context. Therefore, we have modified this discussion to focus on the potential for different P pools (organic/inorganic, particulate/dissolved) to have different cycling in the environment (e.g. dissolved phase will be more reactive/have quicker cycling, solid P may be longer term source of P to the environment) compared to starting vegetation P. Lines 539-552.**

**We also have removed soil from our key words.**

**Reviewer 2:**

**Dear authors,**

This paper is an important contribution to understanding the biogeochemical impacts of fire on phosphorus fate and mobility. The study employs a robust methodological approach, including NMR and XANES analyses, to investigate how burn severity and vegetation type influence phosphorus composition and mobilization. The findings are relevant for understanding post-fire nutrient cycling and have implications for both terrestrial and aquatic ecosystems.

**Response: Thank you for your favorable review of our manuscript.**

**Major Comments:**

1. Providing information on the geological and soil characteristics of the study site, including soil type and pH, would help contextualize phosphorus dynamics. Additionally, details on climate conditions such as rainfall, temperature, and seasonality would improve the interpretation of post-fire phosphorus mobilization and retention. If the bedrock is limestone, it could increase soil calcium

concentrations and pH, reducing bioavailable phosphorus and promoting the formation of Ca-bound phosphorus (Ca-P). This may influence the phosphorus speciation in the original soil, affecting the composition of the phosphorus pool from which the organic samples were taken. If available, correlating soil properties with phosphorus fractions could further strengthen the study.

**Response: This study was designed to examine changes across land cover archetypes across the Pacific Northwest, not a specific site study. This is notably a laboratory experiment and simulates P mobilization by leaching chars with artificial rainwater and does not include study of soils or in-situ leachates from litter/overland flow. We have clarified these points throughout Section 2.1.**

2. **Soil Phosphorus Pools and Pre-Fire Conditions:**

   It would be valuable to include data on the total phosphorus concentration in the soils from which the organic matter was sourced. Since the initial phosphorus pools in the soil can influence post-fire phosphorus mobilization and retention, this information could provide important context for interpreting the results. If this data is available, correlating soil phosphorus status with the observed trends would further strengthen the study.

   **Response: The organic matter burned here was plant material (i.e. litter), not soils. The initial pools of P in the study can be considered that of the unburned plant materials presented throughout the manuscript. We have modified Section 2.1 to increase clarity.**

3. **Use of the Term "Total P Concentration"** (Line 99 and elsewhere):

   The term "total P concentration" is unclear. It would be helpful to clarify whether this refers to the sum of all phosphorus fractions or a specific measurement of P concentration. Consistently defining this term throughout the text would improve clarity.

   **Response: Total P is measure of a P via ICP-OES. We define this upon first usage on line 106 and again in the methods on line 245.**

**Specific Comments:**

- **Line 65:** Expand on the impacts of climate change on fire regimes. The statement that fires are expected to increase in intensity and severity could be

strengthened by elaborating on the mechanisms driving this trend, as climate change is expected to exacerbate fire risk.

**Response: We agree with the reviewer there are several factors exacerbated by a changing climate that are influencing fire regimes in our study region. We have modified lines 69-75 accordingly.**

- **Line 285 (Figure 1):** The distinction between moderate and high temperatures for Douglas fir is unclear. The figure appears to show overlapping temperature ranges - maybe you can think of a different phrasing? Additionally, the lines on the figure should be explained—do they represent the 25th and 75th percentiles? Are they median or mean values?

**Response: Thank you for bringing this point of confusion to our attention. Burn severity was assessed via visual metrics, which includes ash color, degree of charring, and degree of consumption (Parsons et al., 2010). Therefore, burn severity is not just dependent on temperature, but is also impacted by duration of heating and additional fuel loading metrics. We had originally included temperature in our figure axes to increase comparability with temperature-based studies of P chemical changes common in the literature. To address this comment, we moved Table S4 to the main text (now Table 1) to better set up our burn severity definition and details of the burn table experiments studied herein. We also point to the SI figure on burn severity (Figure S1) in the main text (lines 290; 513).**

**Regarding Figure 1, this is a boxplot. We have annotated in the figure legend this detail to reduce possible confusion on interpretation, removed the temperature ranges on the axis, and removed colors from the boxplots for simplicity. We also discuss how the data is presented on lines 276-279. We refer to the table brought into the main text from the SI for sample numbers of each observation and the temperature ranges for each severity (Table 1).**

**References:**

**Parson, Annette, et al. "Field guide for mapping post-fire soil burn severity." *Gen. Tech. Rep. RMRS-GTR-243. Fort Collins, CO: US Department of Agriculture, Forest Service, Rocky Mountain Research Station. 49 p.* 243 (2010).**

- **Line 334 (Figure 2):** The color scheme should be adjusted to better differentiate between organic P and inorganic P. Additionally, the use of "moderate" and

"high" temperatures for Douglas-fir is unclear, as it appears inconsistent with figure 1.

**Response: We have added in hashing onto the colors for organic P in both this figure and figure 3 to increase clarity and removed the temperatures from figure legend.**

- **Line 453 (Figure 3):** Similar to Figure 1, further clarification is needed. What do the dots above the boxes represent? How many samples were measured? The figure suggests that Douglas-fir moderate fire has significantly higher leachable P than low-temperature burns, and that high-temperature burns are significantly higher than moderate burns. For sagebrush, in the lower right panel, the "a" and "a" labels above the boxes indicate significant differences, yet the values appear different. Please review and clarify.

  **Response: We believe this comment is in reference to figure 4, not figure 3. Similar to figure 1, we now clearly state in the figure legend that this is a boxplot. We have also removed the temperature ranges on the axis and removed colors from the boxplots for simplicity. We have clarified the lettering for significant differences in the figure caption (briefly, those with the same lettering above them are not significantly different). We refer to the table brought into the main text from the SI for sample numbers of each observation and the temperature ranges for each severity (Table 1).**

- **Line 455:** The impact of pH on phosphorus solubility is mentioned, but actual pH values are not provided. I suggest including these values in the results.

  **Response: In response to this comment and a similar comment from R1, we have moved Figure S7 from the SI into the main text, now Figure 6.**

Line 470: The proportion of Na-P is relatively high in Douglas-fir. A brief discussion of the potential role of Na-P in post-fire phosphorus dynamics would be useful.

**Response: Our current understanding of P biogeochemistry is largely around Ca-, Al-, and Fe-P given the popularity of sequential chemical fractionation schemes, which infer speciation of these compounds based on solubility (Kruse et al 2015). XANES is increasingly being used to identify P speciation within environmental samples, but many studies still focus on Ca-, Al- and Fe-P compound identification. However, other studies characterizing chars have identified Na-P (i.e., Rose et al 2019) and Li and Brett (2013) have identified sodium tripolyphosphate as having high nutrient uptake and bioavailability**

during bioassay experiments. We have added these viewpoints to lines 499-500; 503-505; 539-552.

---

## Author Response (AR2)

**Public justification (visible to the public if the article is accepted and published)**: Thanks to the authors for their comprehensive responses to the reviewer comments. It is my view that the manuscript is very close to being accepted, but I would like to see a few extra components address and then will be happy to move to publication.

Thank you for the favorable impression of our revisions. We have provided responses to each request of the editor below in blue text.

I see the removal of temperature ranges from the figures relates to responses to reviewer 2. However, I believe the temperature ranges are an important component of the study and I believe the figures benefit from their inclusions – readers want to see the range of these temperatures. My recommendations would be to simply add a better description of why the temperatures overlap (the temperatures are not the treatments, rather the ecosystem types) in the methods. I still believe this is underdone in the methods. While Table 1 provides some of this temperature data, the confusing part that remains is that there is no distinction in temperature between moderate and high temperature burns in Douglas Fir forests. Please better explain how categories of "low", "moderate" and "high" temperature treatments (for lack of a better term) were defined, particularly for Douglas Fir forests, but indeed all the ecosystem types.

We do not use temperature treatments in this study, but rather characterize alterations to the organic matter in terms of burn intensity and resultant burn severity. Burn severity is defined as the effect of fire on the environment, through changes in vegetation and soil after a fire. It is not a measure of fire intensity (the amount of energy released form the fire), nor burn temperature, though it is related to both. For example, you could have a very hot fire that burns through an ecosystem rapidly but produces a low burn severity because it did not actually have time to transform or consume much of the above or belowground biomass during the burn.

We used a field guide for assessing post fire soil burn severities to assess the burn severities of our experimental burns. These are done visually, through assessment of the degree of alteration to the materials burned, after the fire is extinguished. Burn severity was assigned to each solid char samples using visual post-fire field metrics from the descriptions outlined in USFS Patterson et al. 2010 field guide. Burn severity was visually determined based on ash color, degree of consumption, and degree of char on vegetation. Burn severity categories assigned were low, moderate, or high. Control vegetation that was not combusted or burned were

classified as unburned. Assigned severities were recorded for each char sample and photographs exist in our data package, and examples of each are in SI figure 1.

Given these details above, we do not want to propagate any conflation of burn temperature with burn severity. We believe that adding the max temperature ranges back into the figures will decrease clarity on our methodology, possibly enabling incorrect interpretation of our figures. Therefore, we have elected to keep the maximum temperatures off the figures and have clarified the text surrounding these points throughout the manuscript in line with the descriptions above (Lines 118-119, 123-124, 164, 168, 172 in the track changes version of the manuscript).

Lines 60-61: The addition here does not add much value and I would make the argument that it actually detracts from the flow of the study because it is difficult to understand what this information from Ball et al. (2021) actually means based off the sentence as it is written. Please either remove the statement or clarify more clearly what is mean by "impacted 11% of total western US river length in recent years". This statement is not intuitive and contains ambiguities (i.e., "recent years").

This has been deleted as suggested.

Lines 308-310: ">774degC or greater", please remove the redundancy "or greater". As suggested by reviewer 1, please add a comment that these data likely underestimate P releases from higher intensity burns (relating to the volatilization temp of P).

">" has been deleted per suggestion. We added the following sentence immediately after this on lines 302-304 in the tracked changes version of the manuscript: *"Therefore, our study may underestimate P transformations linked to P volatilization from burns that reach higher temperatures than our experimental burns."*

Lines 368: Issues with the axis label overlapping the axis.

We believe that this issue was only in the track changes version of the figure on line 368, as we do not see this issue in the final pdf version or the individually uploaded figure. Please let us know if you are still seeing this issue in the current version of the manuscript and figures.